# STING pathway contributes to Steroid-Hyporesponsive Lung Inflammation in DSS-induced colitis mice model

**Mariam Wed Eladham**[1], **Narjes Saheb Sharif-Askari**[1,2], **Bushra Mdkhana**[1], **Shirin Ali**[1], **Baraa Khalid Salah Al-Sheakly**[1], **Nival Ali**[1], **Balachandar Selvakumar**[1], **Fatemeh Saheb Sharif-Askari**[1,3], **Ibrahim Hachim**[1,2], **Rabih Halwani**[1,4,5,6,7] *

1 Research Institute for Medical and Health Sciences, University of Sharjah, Sharjah, United Arab Emirates, 2 Department of Clinical Sciences, College of Medicine, University of Sharjah, Sharjah, United Arab Emirates, 3 Department of Pharmacy Practice and Pharmacotherapeutics, College of Pharmacy, University of Sharjah, Sharjah, United Arab Emirates, 4 King Saud University Celiac Disease Research Chair, Department of Pediatric, College of Medicine, King Saud University, Riyadh, Saudi Arabia, 5 Health and wellbeing sector, NEOM, Tabuk, Saudi Arabia, 6 Prince Fahad Bin Sultan chair for Biomedical Research, University of Tabuk, Tabuk, Saudi Arabia, 7 College of Medicine, Alfaisal University, Riyadh, Saudi Arabia

* rhalwani@sharjah.ac.ae

## Abstract

### Background

Inflammatory bowel disease (IBD), a chronic inflammation of the gastrointestinal tract, is well recognized for triggering extraintestinal manifestations, including pulmonary complications. Emerging evidence highlights the *gut lung axis* (GLA) as a critical link in respiratory health, where gut dysbiosis and bacterial translocation play a role in systemic and pulmonary inflammation. Despite its clinical relevance, the mechanisms underlying these pulmonary manifestations remain poorly understood. The Stimulator of Interferon Genes (STING) pathway plays a critical role in regulating pulmonary inflammation. However, its precise role in colitis-associated lung inflammation remains unclear and could provide novel insights into the pathogenesis of this condition.

### Methods

This study evaluates the involvement of STING pathway in colitis induced lung tissue inflammation using a dextran sulfate sodium (DSS) murine model of colitis. The effect of STING inhibitor on regulating steroid hypo-responsiveness, particularly the glucocorticoid receptor GR-α/GR-β ratio, is also examined.

### Results

The DSS model induces lung inflammation, characterized by enhanced infiltration of inflammatory cells into lung tissues, increased levels of IL-17, IFN-γ, bacterial DNA,

**Data availability statement:** All relevant data are within the paper and its Supporting Information file.

**Funding:** This research has been supported by Ongoing Research Funding program - Research Chairs(ORF-RC-2026-0500),King Saud University, Riyadh, Saudi Arabia.

**Competing interests:** The authors have declared that no competing interests exist.

while enhancing steroid hypo-responsiveness. The inhibition of STING controls lung inflammation and restores steroid sensitivity to a much higher extent compared to dexamethasone treatment.

## Conclusion

The significant activation of the STING pathway and dysregulation of steroid signature markers in the lungs of DSS-induced colitis mice suggest a novel mechanism by which gut inflammation may propagate to the lungs.

## 1. Introduction

Inflammatory bowel diseases (IBD), including ulcerative colitis (UC) and Crohn's disease (CD), have long been recognized for their primary gastrointestinal manifestations. However, growing evidence shows that IBD also disrupts lung homeostasis and immune responses, reflecting the systemic nature of the disease [1–3]. Pulmonary involvement in IBD is thought to arise through the *gut lung axis (GLA)*, where microbial translocation and systemic immune activation originating in the gut drive inflammatory responses in the lungs [4–6]. Once considered rare, IBD-associated pulmonary disease is an increasingly reported entity [2]. Clinical and experimental studies have demonstrated that patients with IBD can exhibit respiratory symptoms, abnormal pulmonary function tests, and radiologic lung abnormalities even during periods of intestinal remission [7–12]. These findings suggest that pulmonary inflammation in IBD does not strictly parallel intestinal disease activity and may persist despite apparent clinical control of gut inflammation [13,14]. Such observations support the concept that systemic immune dysregulation, rather than local intestinal injury alone, contributes to extraintestinal manifestations of IBD.

Although significant progress has been made in understanding the gut lung crosstalk and its role in lung inflammation [15,16], the specific mechanisms by which gut dysbiosis regulates steroid responsiveness in the lungs remain unexplored. Bacterial DNA and microbial translocation have been recently proposed — by our group and others — as key contributors to lung inflammation [3,15], particularly in colitis models where inflammation originates primarily in the gut. Among the innate immune pathways activated by microbial products, the cyclic GMP–AMP synthase–stimulator of interferon genes (cGAS–STING) pathway has emerged as a key regulator of inflammation [17].

The stimulator of interferon genes (STING), encoded by the *TMEM173* gene, is a transmembrane adaptor protein localized predominantly to the endoplasmic reticulum. It plays a central role in innate immune sensing by mediating downstream signaling in response to cytosolic DNA detected by cyclic GMP–AMP synthase (cGAS) [18]. STING is essential for host defense against pathogens, yet its dysregulation has been implicated in autoimmune and chronic inflammatory diseases, including IBD [19]. In IBD, cytosolic DNA from damaged host cells or microorganisms activates the cGAS–STING signaling cascade, leading to type I interferon and pro-inflammatory

cytokine production [18,20]. Although the role of STING in intestinal inflammation is increasingly recognized, its contribution to IBD-mediated extraintestinal manifestations, such as lung inflammation, remains poorly understood. The GLA may therefore represent a critical axis of immune regulation, with the STING pathway functioning as a shared mediator in gut and lung pathology.

Corticosteroids remain a standard therapy in the management of IBD [21]. Despite their efficacy, 20–40% of patients with CD or UC develop steroid resistance or hyporesponsiveness, particularly during severe or prolonged disease flares [22–25]. Given that intestinal barrier dysfunction and microbial translocation can promote systemic immune activation, it is plausible that mechanisms underlying steroid hyporesponsiveness in the gut may also extend to extraintestinal sites. In this context, pulmonary inflammation associated with IBD may share similar pathways of impaired steroid responsiveness. However, despite increasing recognition of gut–lung immune crosstalk, the potential contribution of steroid resistance to extraintestinal manifestations, particularly in the lung, has not been systematically investigated.

Recent work from our group and others has demonstrated that activation of the STING pathway contributes to steroid resistance in inflammatory lung disease [26], raising the possibility that STING-driven signaling may represent a shared mechanism underlying both intestinal and pulmonary inflammation in IBD. Based on this rationale, we hypothesized that STING activation contributes to colitis-associated lung inflammation and that pharmacological addition of STING inhibitors could attenuate pulmonary pathology and restore steroid responsiveness.

In this study, we therefore investigated the role of STING in a murine model of DSS-induced colitis with a specific focus on lung inflammation as an extraintestinal manifestation. We examined whether pharmacological inhibition of STING modulates pulmonary inflammation and steroid responsiveness, and evaluated its impact on both lung and intestinal pathology. By defining the contribution of STING signaling to gut–lung immune crosstalk, our findings aim to clarify the mechanistic basis of steroid hyporesponsiveness in IBD and identify STING as a potential therapeutic target for managing extraintestinal complications.

## 2. Methods

### 2.1. Animal care, monitoring, and humane endpoints

Experiments were carried out in wild-type female C57BL/6 mice (6–8 weeks) obtained from the University of Sharjah animal facility. Mice were housed under specific pathogen-free conditions in sterile, individually ventilated cages (TECNIPLAST) on a 12-h light/dark cycle. Animals had ad libitum access to a sterile maintenance diet (Altromin 1324 TPF) and sterile distilled water. Autoclaved, dust-free aspen bedding (ABEDD) was provided for environmental enrichment. Mice were monitored at least once daily throughout the experimental period for general health status, activity, and grooming. Body weight, stool consistency, and the presence of rectal bleeding were recorded individually, and a Disease Activity Index (DAI) was calculated to assess colitis severity. Humane endpoints were established prior to study initiation in accordance with the University of Sharjah Animal Care and Use Committee guidelines (Approval No. ACUC-06-02-2024). Mice were euthanized if they exhibited more than 20% body weight loss from baseline, severe rectal bleeding, marked dehydration, lethargy, persistent inability to access food or water, or any other signs of undue suffering. Euthanasia was performed under deep isoflurane anaesthesia followed by cervical dislocation to ensure rapid and humane death. All efforts were made to minimize animal suffering, and survival studies were conducted only to the extent necessary to achieve scientific objectives.

Sample size was determined based on previous studies assessing DSS-induced colitis and lung inflammation in mice, aiming to detect significant differences in histologic and molecular endpoints with 80% power and $\alpha = 0.05$. Experiments were performed in at least three independent biological replicates, with multiple technical replicates where appropriate, to ensure reproducibility and reliability of results.

## 2.2. Experimental colitis-induced lung inflammation with DSS

Gut inflammation was induced by adding 4% weight/volume dextran sulfate sodium (DSS) in drinking water ad libitum for 5 days. Mice were free from DSS for 2 consecutive days until day 7. Mice (35 animals) were randomly assigned to seven groups, each with 5 animals as shown in (Fig 1A). Control mice in this cohort received normal drinking water only, as DSS was administered via drinking water.

In the 15-day treatment cohort, mice were allocated to five groups: control, DSS, DSS-Dex, DSS-H151, and DSS-Dex-H151. All mice in this cohort were sacrificed on day 15. Dexamethasone and H151 were dissolved in PBS and administered intranasally in a total volume of 20 μL per mouse. Control mice in this cohort received intranasal PBS (20 μL) following the same schedule as treatment groups. All comparisons were performed within the same experimental cohort and timepoint.

DSS challenged mice were treated with intranasal Dexamethasone **(0.25 mg/kg)**, intranasal STING inhibitor (H151) **(523.8ng/kg)** or combination treatment. Mice were treated with corresponding treatment on days 9, 11, 13. Control group mice received normal drinking water.

Water and food intake were monitored daily at the group level, while body weight, stool consistency, and the presence of visible blood in the feces were evaluated individually for each mouse throughout the experimental period. The DAI was calculated based on the method as previously established [27], using the formula: total score = (body weight loss + stool consistency + rectal bleeding)/ 3. This provides a composite score to assess disease severity in each mouse. Mice were sacrificed on either day 7 or day 15.

## 2.3. Ethics Statement

All procedures were performed in accordance with the protocols approved by the Animal Care and Use Committee of the University of Sharjah (Approval No. ACUC-06-02-2024). At the end of the experiments, mice were anesthetized with isoflurane inhalation and euthanized by cervical dislocation, with efforts taken to minimize suffering.

## 2.4. Airway Hyperresponsiveness

Airway hyperresponsiveness (AHR) was assessed by measuring the total lung resistance (Rrs, cm $H_2O.s/mL$) using the FlexiVent system (SCIREQ, Montreal, Canada). Mice were anaesthetized with 114.5 mg/kg ketamine and 6.9 mg/kg xylazine administered intraperitoneally and tracheostomized with stainless steel cannula and then nebulized to increasing doses of methylcholine (0–50 mg/ml, MCh, Sigma). Airway resistance was reported as calculated in cm H20/mL using FlexiVent software, version 8.0.4.

## 2.5. Histopathological Scoring

Colon and Lung tissues were fixed in 10% formalin, embedded in paraffin blocks, and were stained with haematoxylin and eosin (H&E). The scoring of inflammation was gathered blindly by two independent investigators. Minor discrepancies in scoring occurred in fewer than 5% of samples and were resolved through joint review and consensus discussion. This approach ensured objectivity, consistency, and reproducibility in the assessment of tissue inflammation and damage.

H&E staining was evaluated and scored based on a modified scoring system described in our previous study [3].

## 2.6. Quantitative real-time polymerase chain reaction (qRT-PCR)

Total RNA was extracted from lung tissues using Tiazol/ chloroform extraction method. RNA was used for cDNA synthesis. The RNA yield was measured by nanodrop2000 spectrophotometer (ThermoScinetific, USA). cDNA was synthesized using RNA using High-Capacity cDNA Reverse Transcription Kit (ThermoScinetific, USA).

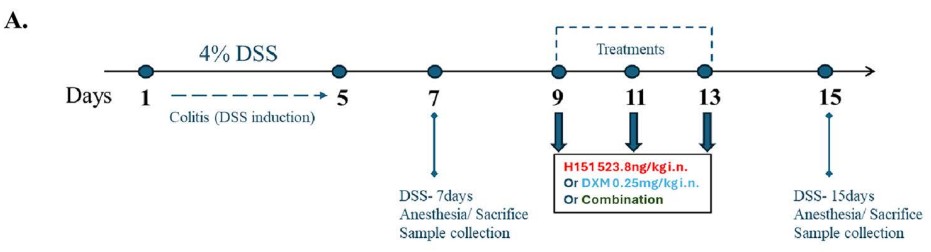

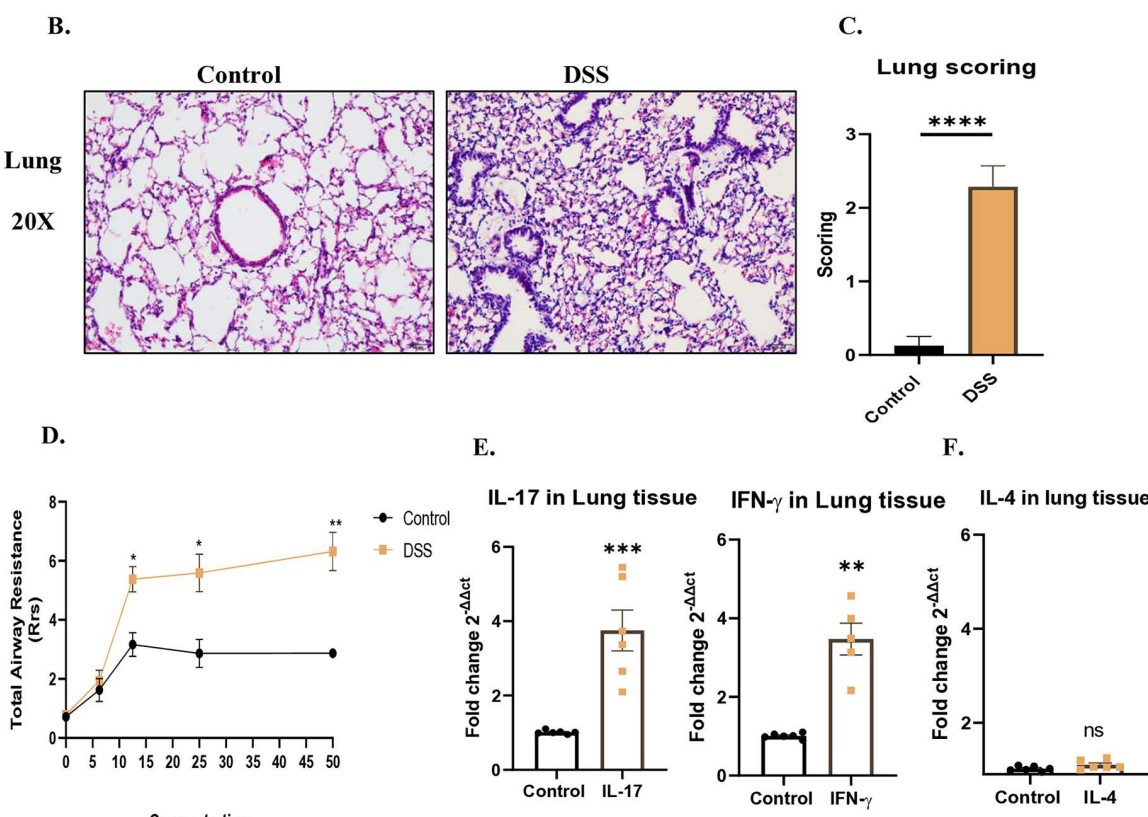

**Fig 1. Histological and molecular assessment of lung inflammation in DSS-induced colitis model.** *Mice were treated with 4% DSS in drinking water for 5 days, followed by 2 days of regular water, to induce acute colitis. (A) Schematic timeline of DSS administration and assessments. (B) Representative H&E-stained lung sections illustrating inflammatory changes, including congestion, alveolar haemorrhage, and leukocyte infiltration in DSS-treated mice. (C) Quantification of lung inflammation severity scores showed significantly higher inflammation in DSS-treated mice than in controls. (D) Total airway resistance was significantly increased following DSS challenge, indicating compromised lung function. (E-F) Gene expression analysis of inflammatory mediators in lung tissue revealed significant upregulation of Il-17 and Ifn-γ in DSS-treated mice, while Il-4 expression remained unchanged. Samples were obtained from control mice and DSS treated mice. Data represent n = 5 mice per group Data represent mean ± SEM. Statistical analysis was performed using one-way ANOVA followed by a post hoc Bonferroni test for multiple comparisons; \*P < 0.05, \*\*P < 0.01, \*\*\*P < 0.001, \*\*\*\* P < 0.0001 compared to control mice.*

Real-Time PCR was performed to quantify the expression level of Cytokines (*Il-4, Il-17, Ifn-γ*), sting pathway genes (*Sting, Tbk1, Irf3, Ifn-β*), and steroid-responsive signature markers (*Grα, Grβ, Hdac-2*), *β-actin* was used as housekeeping gene. The primers are listed below in (Table 1) Real-time PCR was conducted on Quant Studio 5 (Thermo Fisher) using 5x HOT FIREPol® EvaGreen® qPCR Supermix (Solis BioDyne).

All samples were amplified in triplicates. The average threshold cycle (Ct) values for the genes were obtained from each reaction, and the expression was quantified using the $2^{(-\Delta\Delta C(T))}$ relative method.

**2.6.1. Expression of total bacterial 16S by RT–qPCR.** DNase-treated total RNA was reverse-transcribed into cDNA, and qRT PCR was performed to detect total bacterial 16S as a measure of bacterial RNA presence in lung tissue. Appropriate negative extraction controls and no-template controls were included to exclude genomic DNA amplification.

## 2.7. Western Blot

Mice lung tissue lysates were prepared in 1X RIPA lysis buffer and cocktail of inhibitors. BCA Protein Assay Kit (Thermo Scientific) was used to determine the quantity of proteins. Cell lysates were diluted in 1X Laemelli's buffer solution for 5 min at 95 ◦C. Total protein was transferred to nitrocellulose membranes after separation on SDS-polyacrylamide gels. 5% BSA in Tris-Buffered saline with Tween (TBST) was used for membrane blocking. Then, membranes were incubated with primary antibodies, anti-phospho-STING (cat. #72971, 1:1000, mouse), anti-STING (cat. #50494, 1:1000), anti-phospho-TBK1/NAK (cat. #5483,1:1000), anti-TBK1/NAK (cat. #3504, 1:1000), anti-phospho-IRF3 (cat. #29047, 1:1000), anti-IRF3 (cat. #4302, 1:1000), and β actin (cat. #4970, 1:1000) (Cell Signaling Technologies, Danvers, MA), and anti-glucocorticoid receptor α (PA1–516), anti-glucocorticoid receptor β (PA3–514) (Thermo Fisher Scientific, Waltham, MA) at 4◦C overnight. The membranes were then probed with anti-rabbit IgG, horseradish peroxidase-conjugated secondary antibody (cat. #7074S, 1:1000; Cell Signaling Technologies) for 1 h at room temperature. Protein bands were visualized using the Sapphire™ NIR-Q Biomolecular Imager (Azure Biosystems, Dublin, US) and quantified with ImageJ software.

## 2.8. Statistical analysis

All data are presented as mean±standard error of the mean (SEM). For in-dependent two group comparisons, an unpaired independent Student's t-test was used, while one-way ANOVA followed by Bonferroni's post hoc test was applied

**Table 1. List of mice primer sequences used in qRT-PCR.**

| Genes | Forward Primer Sequence (5′-3′) | Reverse Primer Sequence (5′-3′) |
|---|---|---|
| *Sting* | CCTAGCCTCGCACGAACTTG | CGCACAGCCTTCCAGTAGC |
| *Tbk1* | GCTCGAGAGCTGGAGGACGATG | CCACAGATCAACGGTAGCCCC |
| *Irf3* | TGGACGAGAGCCGAACGAGGTT | TGTAGGCACCACTGGCTTCTGC |
| *Ifn-β* | CCTCACCTACAGGGCGGACTTC | TCATTCCACCCAGTGCTGGAGA |
| *Il-4* | GCATTTTGAACGAGGTCACAGG | CTCTCTGTGGTGTTCTTCGTTG |
| *Ifn-γ* | TCAACAACCCACAGGTCCAGCA | TCAGCAGCGACTCCTTTTCCG |
| *Il-17a* | ACCGCAATGAAGACCCTGAT | TCCCTCCGCATTGACACA |
| *Grα** | AAAGAGCTAGGAAAAGCCATTGTC | TCAGCTAACATCTCTGGGAATTCA |
| *Grβ** | AAAGAGCTAGGAAAAGCCATTGTC | CTGTCTTTGGGCTTTTGAGATAGG |
| *Hdac-2* | GGAGGAGGCTACACAATCCG | TCTGGAGTGTTCTGGTTTGTCA |
| Total bacterial *16s* | TCCTACGGGAGGCAGCAGT | GGACTACCAGGGTATCTAATCCTGTT |
| *βactin* | CATTGCTGACAGGATGCAGAAGG | TGCTGGAAGGTGGACAGTGAGG |

*Grα and Grβ* * used from [28].

for multiple comparisons. A p-value of less than 0.05 was considered statistically significant. Data analysis was performed using GraphPad Prism version 8.4.2 (GraphPad Software, Inc., La Jolla, CA, USA).

## 3. Results

### 3.1. Characterization of the experimental model of DSS-induced colitis

To explore the link between colitis and lung inflammation, we used an acute model of DSS-induced colitis as previously described [16]. Briefly, mice were subjected to DSS at concentration 4% in drinking water for 5 days followed by drinking water for 2 days prior to assessment (Fig 1A). The progression of colitis was evaluated through colon length, body weight monitoring and DAI scoring. Mice subjected to DSS exhibited significant weight loss starting from day 4, which continued to worsen over days 5–7 (means ± SEM (original weight-g): control, 20.92 ± 0.0274; DSS, 16.36 ± 0.4026; $P < 0.0001$) (S1A Fig). Additionally, DSS challenged mice had a significant reduction in colon length (means ± SEM (original colon length-cm): control, 10.22 ± 0.3736; DSS, 5.850 ± 0.2432; $P < 0.0001$) (S1B Fig), and increase in colitis severity, as measured by DAI scores, compared to the control group ($P < 0.001$) (S1c Fig). Histological analysis of colon tissue confirmed increased colonic inflammation in DSS-treated mice, as evidenced by significant transmural inflammation. This was reflected in the elevated inflammation scores, showing a pronounced crypt damage, immune cells infiltration into the lamina propria, as well as alterations in epithelial and mucosal structures, compared to non-DSS-treated controls by day 7 post-DSS challenge (means ± SEM (arbitrary units): control, 0.24 ± 0.20; DSS, 3.143 ± 0.3401; $P < 0.0001$) (S1D-S1E Fig). Collectively, these changes are hallmarks of severe colonic injury and indicate a robust inflammatory response following DSS exposure.

### 3.2. DSS-induced colitis results in lung inflammation

Previous studies have suggested that lung inflammation is promoted in the DSS-induced colitis model via the gut-lung axis [16,29]. To investigate that, we first performed histological analysis of H&E-stained lung sections from the DSS-induced colitis model. Lung inflammation was characterized by congestion, alveolar haemorrhage and leukocyte infiltration around the airway vasculature and parenchyma in DSS-treated mice compared to controls (means ± SEM (arbitrary units): control, 0.1250 ± 0.120; DSS, 2.375 ± 0.2631; $P < 0.0001$) (Fig 1B-1C). Interestingly, total airway resistance also significantly increased following DSS challenge when compared to control (Fig 1D).

One of the hallmarks of colitis-induced mouse model is the increased levels of IL-17 and IFN-γ [30]. Therefore, the expression of these mediators was subsequently examined in lung tissues. Gene expression analysis showed significant upregulation of *Il-17* (means ± SEM (fold change): control, 1.012 ± 0.02023; DSS, 3.751 ± 0.5501; $P = 0.0006$) and *Ifn-γ* (means ± SEM (fold change): control, 1.007 ± 0.02753; DSS, 3.473 ± 0.4061; $P < 0.0001$) in DSS-treated mice compared to healthy controls (Fig 1E). However, *Il-4* was not significantly changed in lung tissues of DSS treated mice ($P = 0.1002$) (Fig 1F).

### 3.3. Colitis drives lung inflammation via STING pathway activation

Previous studies have found that microbial factors play a key role in lung inflammation associated with colitis [16]. Our study found a significant increase in total bacterial DNA in the lungs of DSS-treated mice compared to controls ($P < 0.0001$) (Fig 2A), suggesting potential bacterial translocation from the gut to the lungs. In our study, the increase in bacterial DNA in the lung appeared to be a driving factor for the activation of the DNA-sensing STING pathway, thereby contributing to the inflammatory response in the lung.

Given the crucial role of the STING pathway in detecting bacterial DNA, we measured gene and protein expression levels of key STING pathway components in mice lung tissues of DSS induced colitis. Remarkably, gene expression analysis showed a significant increase in mRNA expression levels of STING pathway components, including *Sting* (means ± SEM (fold change): control, 0.8982 ± 0.05452; DSS, 2.747 ± 0.5301; $P = 0.0056$), *Tbk-1*

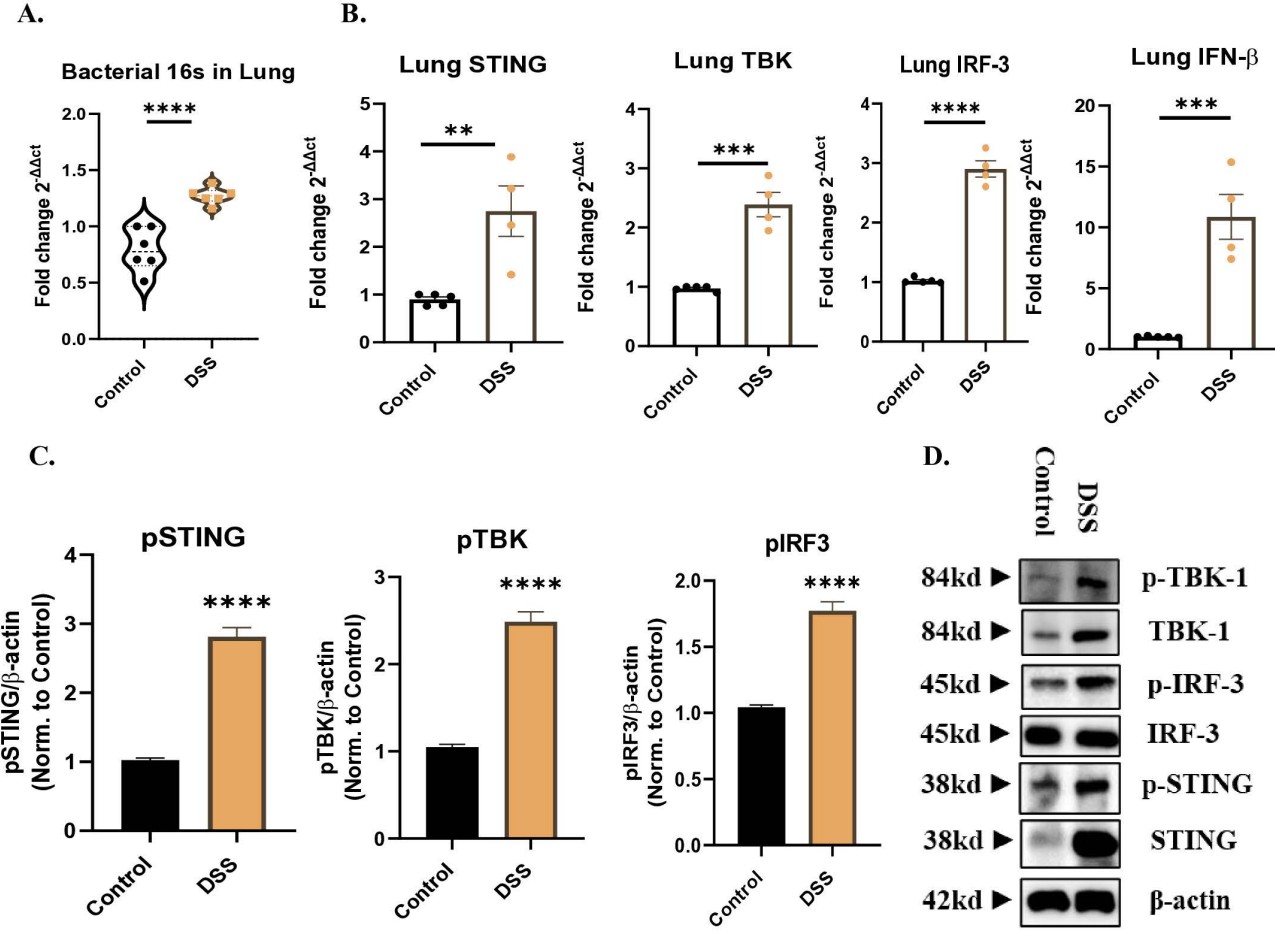

**Fig 2. Role of microbial DNA and STING pathway activation in DSS-induced lung inflammation.** (A) Bacterial 16S expression was measured *in mouse lungs by qPCR of DSS-treated mice. (B) Gene expression analysis of STING pathway components revealed marked increases in Sting, Tbk-1, Irf-3, and IFN-β mRNA levels in DSS-treated lung tissues. Densitometric analyses (C) and western blots (D) demonstrated elevated activation of STING pathway proteins, including phosphorylated STING, TBK1, and IRF3 in DSS-treated mice, confirming STING pathway activation. Samples were obtained from control mice and DSS treated mice. Data represent n = 5 mice per group. Data are presented as mean ± SEM, with statistical analysis performed by unpaired one-way ANOVA followed by a post hoc Bonferroni test for multiple comparisons; *P < 0.05, **P < 0.01, ***P < 0.001, **** P < 0.0001 compared to controls.*

(means ± SEM (fold change): control, 0.9718 ± 0.01963; DSS, 2.388 ± 0.2057; *P* = 0.0001), *Irf-3* (means ± SEM (fold change): control, 1.022 ± 0.02010; DSS, 2.903 ± 0.1369; *P* < 0.0001) and *IFN-β* (means ± SEM (fold change): control, 1.013 ± 0.02320; DSS, 10.86 ± 1.840; *P* = 0.0005), in lung tissues of DSS-induced colitis mice compared to non-colitis control mice (Fig 2B). In alignment with the mRNA findings, STING pathway was significantly activated as appeared in the increased phosphorylation of the pathway proteins, pSTING (*P* < 0.0001), pTBK1 (*P* < 0.0001), and pIRF3 (*P* < 0.0001) (Fig 2C-2D).

Taken together, these data suggested that DSS-induced colitis is associated with lung inflammation, possibly triggered by bacterial DNA and resulting in the upregulation of key components of the innate immune response, mainly STING pathway. This supports the notion that gut-derived microbial signals can directly influence pulmonary immune responses, emphasizing the gut-lung axis in the pathology of inflammatory lung conditions.

 

## 3.4. DSS-induced colitis dysregulates steroid signature markers in mouse lungs

Given the established role of glucocorticoids in modulating inflammatory responses in lung inflammation [31] and in colitis [23], we aimed to explore how DSS induced colitis affects response to steroids in lung tissue of these mice. Glucocorticoid responsiveness was assessed by evaluating the levels of HDAC2 and the GRα/GRβ ratio [32]. GRα is the functionally active glucocorticoid receptor isoform, whereas GRβ acts as a dominant negative regulator, interfering with GRα activity. HDAC2 is a key epigenetic regulator required for glucocorticoid-mediated anti-inflammatory effects, and its reduced expression has been linked to steroid resistance. Together, these biomarkers provide insight into the molecular basis of glucocorticoid responsiveness in our model. We examined the dysregulation of the GRα/GRβ ratio and HDAC2 expression at both mRNA and protein levels in mice lung tissues. Interestingly, DSS mice showed a significant decrease in *Grα* (means ± SEM (fold change): control, 0.8982 ± 0.05452; DSS, 0.5050 ± 0.04550; $P = 0.0005$), and significant increase in *Grβ* (means ± SEM (fold change): control, 1.034 ± 0.01827; DSS, 1.313 ± 0.07041; $P = 0.0050$), with a total reduction in GRα/GRβ ratio (means ± SEM (fold change): control, 0.9386 ± 0.02811; DSS, 0.4384 ± 0.05900; $P < 0.0001$) at mRNA levels (Fig 3A). Additionally, *Hdac2* was significantly reduced in DSS mice compared to controls (means ± SEM (fold change): control, 1.046 ± 0.09173; DSS, 0.5522 ± 0.1076; $P = 0.0082$) (Fig 3A). This was confirmed at protein levels confirming dysregulation in steroid responsiveness biomarkers in lungs of DSS mice group (Fig 3B-3C).

## 3.5. Inhibition of STING pathway reduces mouse lung inflammation and restores steroid sensitivity following DSS challenge

To confirm the role of the STING pathway in DSS-induced lung inflammation, we used STING inhibitor H151, a potent, irreversible, and selective inhibitor of STING [33]. Although treating mice with DSS for 7 days was correlated with significant upregulation in STING pathway and dysregulated steroid markers in lungs, the 14-day model showed more pronounced and sustained inflammation. Histological examination of colonic sections revealed that the 14-day model led to greater tissue damage, epithelial disruption, crypt architectural distortion, and inflammatory cell infiltration (S2A-S2B Figs). Treatment with either dexamethasone or H151 partially ameliorated these pathological features; however, combined treatment with DSS-Dex -H151 resulted in the most pronounced improvement, with preservation of epithelial architecture and reduced inflammatory infiltrates. Consistent with these findings, DSS exposure induced marked body weight loss over the disease course, whereas treatment with dexamethasone or H151 partially mitigated weight loss (S2C Fig). Mice receiving combined DSS-Dex-H151 exhibited the greatest preservation of body weight. Similarly, DSS treatment resulted in significant colon shortening, which was significantly attenuated by both dexamethasone ($P < 0.001$) and H151 ($P < 0.0001$) treatment, with the most pronounced restoration observed in the combination group ($P < 0.01$) (S2D-S2E Figs). Similarly, histological analysis revealed that the 14-day model led to greater tissue damage, including more severe immune cell infiltration and epithelial disruption in the lungs, which better mimic the chronic nature of inflammatory processes seen in human disease. Mice treated with 4% DSS in their drinking water for 7 days, followed by treatment with H151 or dexamethasone three times a week until day 13 as shown in (Fig 1A). Lung inflammation was evaluated by histological examination of lung tissue sections. Inhibition of STING pathway alleviated pathological changes significantly ($P < 0.0001$) in mouse lung tissues, whereas dexamethasone treatment had minimal effect ($P < 0.05$) when compared to DSS mice (Fig 4A-4B). Importantly, lung pathology was significantly improved in DSS-H151–treated mice compared to DSS-Dex–treated mice ($P < 0.001$), as evidenced by reduced inflammatory cell infiltration and decreased alveolar wall thickening, indicating superior efficacy of STING inhibition over corticosteroid treatment in this model. We next proposed using dexamethasone in combination with H151. Lung inflammation was greatly improved upon dexamethasone combination with H151 when compared to DSS-H151–treated mice ($P < 0.05$) and DSS-Dex–treated mice ($P < 0.0001$).

Consistently, the mRNA expression of *Sting* (means ± SEM (fold change): DSS, 2.997 ± 0.3498; DSS-H151, 1.580 ± 0.1934; $P = 0.0012$), *Tbk1* (means ± SEM (fold change): DSS, 2.388 ± 0.2057; DSS-H151, 1.350 ± 0.06455;

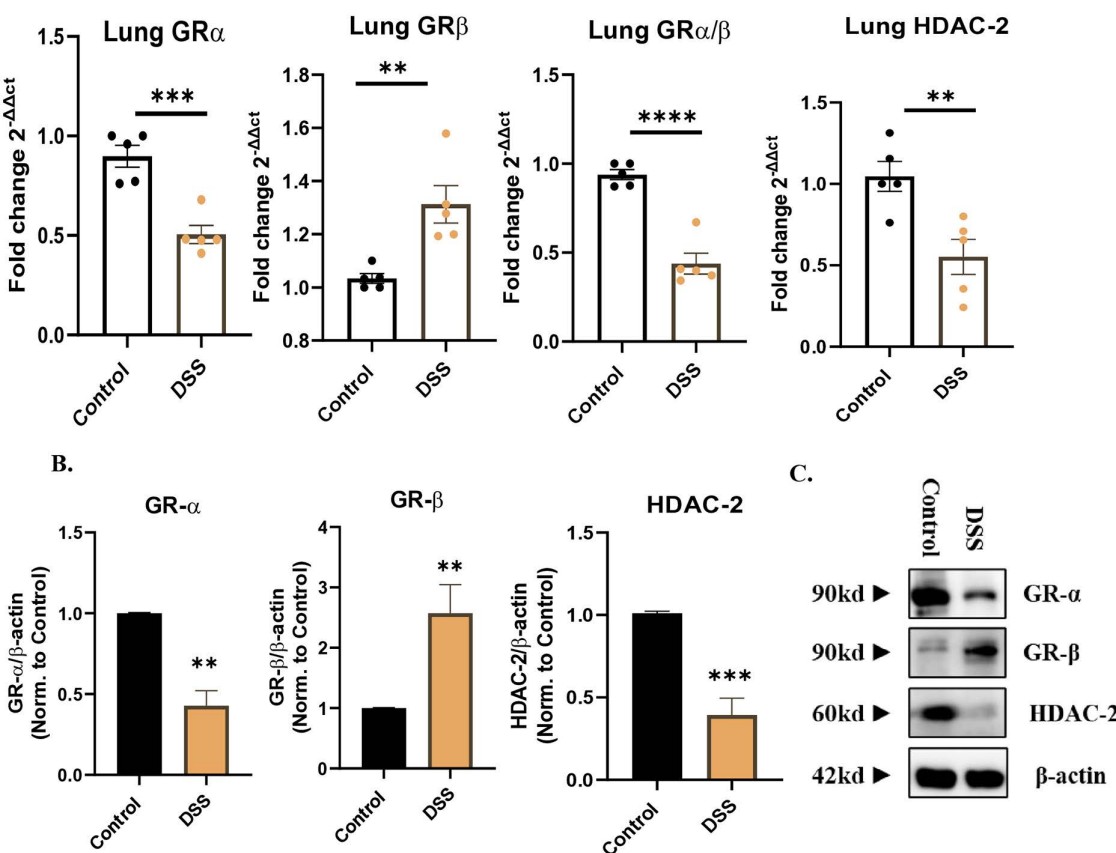

**Fig 3. Dysregulation of glucocorticoid signature markers in lungs of DSS-induced colitis mice. (A)** Representative mRNA analysis of Grα, Grβ, GRα/GRβ ratio, and HDAC2 levels in DSS mice relative to controls, indicating impaired glucocorticoid signalling. **(B-C)** Protein analyses confirmed these findings, showing consistent dysregulation at the protein level. Samples were obtained from control mice and DSS treated mice. Data represent n = 5 mice per group. Results are presented as mean ± SEM, with statistical analysis performed by one-way ANOVA followed by a post hoc Bonferroni test for multiple comparisons; *P < 0.05, **P < 0.01, ***P < 0.001, **** P < 0.0001 relative to control mice.

$P < 0.0001$), *Irf3* (means ± SEM (fold change): DSS, 2.903 ± 0.1369; DSS-H151, 1.250 ± 0.1041; $P < 0.0001$) and *IFN-β* (means ± SEM (fold change): DSS, 7.862 ± 0.2793; DSS-H151, 2.100 ± 0.2799; $P < 0.0001$) was significantly decreased in DSS-mice treated with H151 (means ± SEM (fold change): DSS, 2.997 ± 0.3498; DSS-H151, 1.580 ± 0.1934; $P < 0.0001$), while dexamethasone treatment resulted in a modest, statistically significant decrease compared to the DSS group (means ± SEM (fold change): DSS, 2.997 ± 0.3498; DSS-Dex, 2.042 ± 0.1800; $P = 0.02$), (means ± SEM (fold change): DSS, 2.388 ± 0.2057; DSS-Dex, 1.825 ± 0.04787; $P = 0.0119$), (means ± SEM (fold change): DSS, 2.903 ± 0.1369; DSS-Dex, 2.225 ± 0.1750; $P < 0.0001$), (means ± SEM (fold change): DSS, 7.862 ± 0.2793; DSS-Dex, 6.300 ± 0.4601; $P = 0.0107$) on these genes respectively (S3A Fig). Interestingly, the gene expression of these markers was significantly reduced upon combination treatment relative to dexamethasone ($P < 0.0001$) and H151 ($P < 0.05$) monotherapy.

In parallel, protein expression and activation levels of STING pathway were assessed using western blot analysis. Interestingly, the activation of STING, TBK1, and IRF3 as evident by their phosphorylation was significantly increased

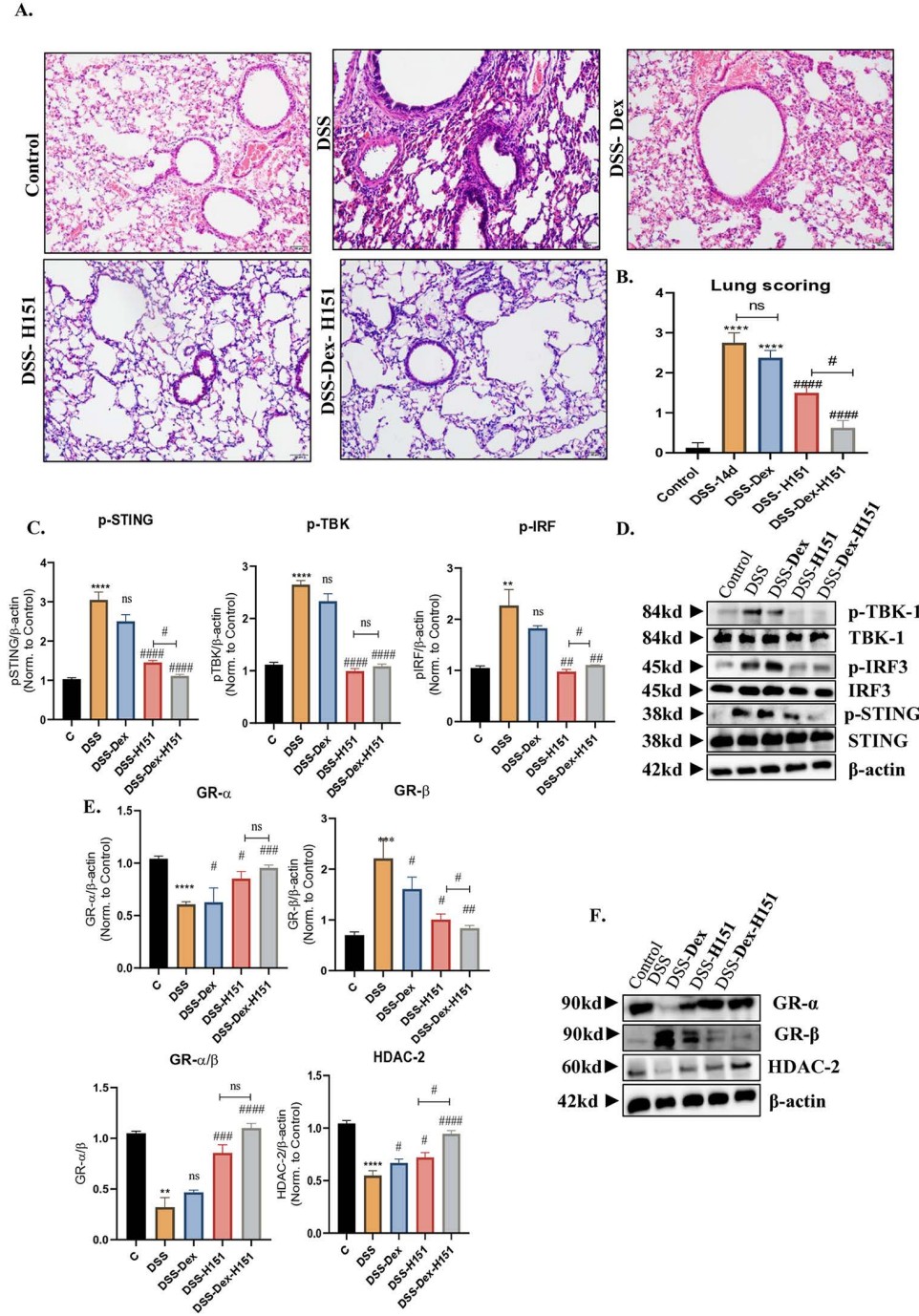

**Fig 4. Role of STING inhibition in modulating DSS-induced lung inflammation and steroid responsiveness.** Mice received 4% DSS in drinking water for 7 days, followed by treatment with the STING inhibitor H151 or dexamethasone on days 9, 11, and 13. **(A, B)** Representative histological sections of lung tissues stained with H&E showing reduced alveolar inflammation, immune cell infiltration, and alveolar wall thickening in DSS mice partially improved by dexamethasone, markedly attenuated by H151, and most effectively resolved by combined dexamethasone and H151 treatment. Densitometric analyses **(C)** and western blots **(D)** of phosphorylated STING, TBK1, and IRF3 protein levels in lung tissue. H151 significantly reduced phosphorylation of STING, TBK1, and IRF3, whereas dexamethasone produced only a modest reduction. Importantly, combined dexamethasone and H151 treatment resulted in significantly greater suppression of STING pathway activation compared with dexamethasone alone. **(E, F)** Representative western blot showing the expression of GRα, GRβ and HDAC2 in lung tissues. Samples were obtained from control mice and DSS treated mice with or

without treatment using Dexamethasone, Sting inhibitor H151 or combination. Data represents n = 5 mice per group. Full blots are supplemented in S1 Raw images. Results are presented as mean ± SEM. Statistical analysis was performed using one-way ANOVA followed by a post hoc Bonferroni test for multiple comparisons: *$P < 0.05$, **$P < 0.01$, ***$P < 0.001$, ****$P < 0.0001$ compared to control mice. #$P < 0.05$, ##$P < 0.01$, ###$P < 0.001$, #### $P < 0.0001$ compared to DSS mice.

in the DSS group ($P < 0.0001$), ($P < 0.0001$), ($P < 0.01$)) and significantly reduced upon H151 treatment ($P < 0.0001$), ($P < 0.0001$), ($P < 0.01$) respectively when compared to DSS. Notably, activation of these signalling proteins remained significantly higher in DSS-Dex mice relative to DSS-H151 mice (Fig 4C–4D), confirming that STING inhibition more effectively suppresses downstream signalling than corticosteroid treatment. Combined treatment further suppressed STING pathway activation, with significantly lower phosphorylation of STING and IRF3 compared to DSS-Dex ($P < 0.001$) and DSS-H151 mice ($P < 0.05$) (Fig 4C–4D).

To further understand the underlying mechanism of steroid hypo-responsiveness colitis induced lung inflammation, we further analysed the steroid signature markers in lung tissues upon treatment with dexamethasone, H151, and combination. Our data showed that dexamethasone mildly influenced the GRα/GRβ ratio, and HDAC2 at mRNA (S3B Fig) and protein levels (Fig 4E, 4F). In contrast, STING inhibition significantly restored GRα/GRβ ratio ($P = 0.0002$) and HDAC2 ($P = 0.0003$) expression with levels significantly higher than those observed in DSS-Dex–treated mice. Moreover, the dexamethasone–H151 combination significantly restored the GRα/GRβ ratio at mRNA level (S3 Fig) with no significant difference at protein levels (Fig 4E-4F) relative to dexamethasone or H151 alone, indicating re-sensitization to corticosteroid signalling.

These results suggested that targeting STING pathway can effectively enhance steroid responsiveness in this model. These findings highlight the potential of STING inhibition as a novel therapeutic approach for overcoming steroid resistance in conditions involved gut-lung axis dysregulation.

## 4. Discussion

In our recently published study, we demonstrated that gut microbial translocation and immune cell mis-homing contribute to lung inflammation in DSS-induced colitis [3]. Building on these findings, the current study shows that DSS-induced colitis not only triggers systemic inflammation but also promotes lung inflammatory responses characterized by elevated IL-17 and IFN-γ, increased bacterial DNA in the lungs, and activation of the STING pathway. changes were accompanied by dysregulation of steroid-responsive markers, including an altered GR isoform ratio and reduced HDAC2 expression, indicating the development of steroid-hyporesponsive lung inflammation. Together, these studies provide complementary evidence supporting the hypothesis that gut inflammation can influence distal organs such as the lungs, potentially contributing to steroid-hyporesponsive inflammation.

In DSS mice model, colonic inflammation induced increased lung inflammation and total airway resistance, consistent with clinical and experimental evidence linking intestinal inflammation to respiratory complications in IBD [15,16]. Colitis models often shift inflammation toward a Th1 and Th17 immune response, mainly through the upregulation of IL-17 and IFN-γ [34]. Both cytokines have been implicated in the pathogenesis of colitis as well as lung inflammation [35–37], suggesting a shared inflammatory milieu between the two tissues. Particularly, IL-17- driven inflammation has been associated with severe steroid hyporesponsive asthma and dysregulation of steroid response markers by our group and others [32,38]. Gut-lung crosstalk could explain the shared steroid-hyporesponsive inflammation and neutrophilic inflammation observed in both IBD and severe asthma [39–41].

Consistent with previous reports, we detected increased levels of Total bacterial 16S in the lungs of DSS-treated mice, suggesting systemic dissemination of microbial products [15,16]. Such bacterial translocation is consistent with the well-documented gut barrier disruption seen in the DSS model, which leads to gut leakage and allows microbial products to enter systemic circulation [42]. This finding aligns with our recent demonstration of bacterial leakage and aberrant

expression of gut-homing receptors in the circulation and lungs of DSS-treated mice [3]. While these finding aligns with the hypothesis that bacterial translocation may contribute to pulmonary inflammation, a direct causal relationship between bacteria and STING activation cannot be established in the current study. Future experiments using germ-free mice or antibiotic-treated models will be necessary to directly test this connection.

DNA sensing pathways represent a potential mechanistic link between microbial products and lung inflammation. Among these, the cGAS–STING pathway and the AIM2 inflammasome are key mediators of cytosolic DNA recognition. While activation of the NLRP3 inflammasome has been well-documented as a key mediator of intestinal and pulmonary inflammation in colitis, the specific contribution of the STING pathway to colitis-associated extraintestinal manifestations remains poorly defined [16]. Our findings demonstrate STING pathway activation in the lungs of DSS-treated mice, suggesting a potential role for this pathway in mediating gut–lung immune crosstalk.

Glucocorticoid resistance remains a clinical challenge in IBD, limiting therapeutic efficacy and often necessitating biologics or surgery [22,23]. In the present study, STING pathway activation in the lungs correlated with dysregulation of steroid-responsive markers, including a decreased GRα/GRβ ratio and reduced HDAC2 expression. These changes are consistent with previous reports linking STING elevation to steroid-hyporesponsive inflammatory phenotype These alterations mirror established mechanisms of glucocorticoid resistance reported in both IBD and severe asthma and align with previous studies linking elevated STING signalling to steroid hyporesponsiveness in pulmonary diseases [26,43–46]. Together, the observed alterations in steroid-responsive markers further support the contribution of STING activation to glucocorticoid resistance in colitis-associated lung inflammation, however, the precise molecular mechanisms linking STING signalling to GR isoform balance and HDAC2 regulation remain to be elucidated.

Emerging clinical evidence suggests that the cGAS–STING pathway is relevant to human IBD pathophysiology. Analyses of intestinal tissues from patients with UC and CD indicate increased expression of STING and downstream signalling components compared with healthy controls, and activation of STING-TBK1-IRF3 correlates with disease severity in human biopsies, supporting its involvement in human intestinal inflammation [47]. Additionally, studies examining human CD4$^+$T cells from IBD patients have demonstrated enhanced STING expression and functional effects on cytokine production [48]. Although the clinical literature remains limited and sometimes conflicting regarding whether STING activation is protective or pathogenic, these observations underscore the translational relevance of investigating STING signalling in both intestinal and extraintestinal inflammatory processes in IBD, warranting further clinical research to clarify its role and therapeutic potential.

Finally, our findings indicate that pharmacological inhibition of STING partially restores GRα/GRβ balance and HDAC2 expression while alleviating lung inflammation (Fig 4). These data suggest the potential of STING inhibitors as therapeutic agents to improve steroid responsiveness. Nevertheless, several limitations should be acknowledged. Although intranasal administration of dexamethasone and the STING inhibitor H-151 was performed under sedation to favor pulmonary delivery, partial systemic absorption or gastrointestinal exposure cannot be fully excluded. Consequently, some observed effects may reflect indirect systemic or gut-mediated influences rather than exclusively local lung actions.

In conclusion, this study provides evidence that gut inflammation can influence distal organs such as the lungs and highlight the STING pathway as a potential modulator of steroid-responsive inflammation. While our findings demonstrate that STING inhibition improves steroid responsiveness in colitis-induced lung inflammation, a direct causal link between microbial translocation and STING activation remains to be established. Future studies employing targeted genetic manipulation of STING pathway components, detailed downstream signalling analyses, and germ-free or antibiotic-treated models will be essential to fully define this regulatory axis and its therapeutic potential in shared steroid-resistant inflammatory diseases such as IBD and severe asthma.

## Supporting information

**S1 Fig. Characterization of DSS-induced colitis model and progression assessment.** Mice were treated with 4% DSS in drinking water for 5 days, followed by 2 days of regular water, to induce acute colitis. (A) Body weight loss was observed in DSS-treated mice starting from day 4 and progressively worsened by day 7. (B) Colon length was significantly shorter in DSS-treated mice compared to controls. (C) Disease Activity Index (DAI) scores indicated an increase in colitis severity in DSS-treated mice. (D-E) Histological scoring of colonic sections showed severe colonic inflammation, with transmural infiltration, crypt damage, and disrupted epithelial and mucosal structures in DSS-treated mice compared to controls. Samples were obtained from control mice and DSS treated mice. Data represent n = 5 mice per group. Results are presented as mean ± SEM, with statistical analysis performed by one-way ANOVA followed by a post hoc Bonferroni test for multiple comparisons; *$P < 0.05$, **$P < 0.01$, ***$P < 0.001$, **** $P < 0.0001$ compared to control mice.
(PDF)

**S2 Fig. STING inhibition enhances dexamethasone-mediated protection against DSS-induced colitis.** Mice received 4% DSS in drinking water for 7 days, followed by treatment with the STING inhibitor H151, dexamethasone or combination on days 9, 11, and 13. (A-B) Representative H&E-stained colon sections demonstrating severe colitis in DSS-treated mice, partially improved by dexamethasone, markedly attenuated by H151, and most effectively resolved by combined dexamethasone and H151 treatment. (C) Body weight changes expressed as percentage of initial weight. DSS induced progressive weight loss, which was modestly improved by dexamethasone, significantly attenuated by H151, and most effectively prevented by combination therapy. (D) Colon length at sacrifice. DSS caused significant colon shortening; this was partially restored by dexamethasone, more effectively preserved by H151, and maximally restored by combination treatment. Results are presented as mean ± SEM. Statistical analysis was performed using one-way ANOVA followed by a post hoc Bonferroni test for multiple comparisons: *$P < 0.05$, **$P < 0.01$, ***$P < 0.001$, ****$P < 0.0001$ compared to control mice. #$P < 0.05$, ##$P < 0.01$, ###$P < 0.001$, #### $P < 0.0001$ compared to DSS mice.
(PDF)

**S3 Fig. STING inhibition suppresses STING pathway activation and restores steroid-responsive markers in the lungs of DSS-treated mice.** (A) mRNA expression of key STING pathway genes (Sting, Tbk1, Irf3, and Ifn-β) measured by qPCR. H151 significantly downregulated these markers compared to the DSS group, with minor effects from dexamethasone. (B) Representative mRNA expression of GRα, GRβ and HDAC2 in lung tissues. Samples were obtained from control mice and DSS treated mice with or without treatment using Dexamethasone or Sting inhibitor H151 or combination. Data represent n = 5 mice per group. Results are presented as mean ± SEM. Statistical analysis was performed using one-way ANOVA followed by a post hoc Bonferroni test for multiple comparisons: *$P < 0.05$, **$P < 0.01$, ***$P < 0.001$, ****$P < 0.0001$ compared to control mice. #$P < 0.05$, ##$P < 0.01$, ###$P < 0.001$, #### $P < 0.0001$ compared to DSS mice.
(PDF)

**S1 Raw images. Full-length, Western blot images corresponding to the immunoblots presented in Figure 4.**
(PDF)

## Author contributions

**Data curation:** Mariam Web Eladham.

**Formal analysis:** Mariam Web Eladham, Ibrahim Hachim.

**Investigation:** Rabih Halwani.

**Methodology:** Mariam Web Eladham, Bushra Mdkhana, Shirin Ali, Baraa Khalid Salah Al-Sheakly, Nival Ali, Balachandar Selvakumar.

**Supervision:** Narjes Saheb Sharif-Askari, Rabih Halwani.

**Validation:** Mariam Web Eladham.

**Writing – original draft:** Mariam Web Eladham, Bushra Mdkhana.

**Writing – review & editing:** Mariam Web Eladham, Narjes Saheb Sharif-Askari, Bushra Mdkhana, Shirin Ali, Baraa Khalid Salah Al-Sheakly, Balachandar Selvakumar, Fatemeh Saheb Sharif-Askari, Ibrahim Hachim.

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
