## [Decision Letter · Decision Letter 0]

17 Nov 2025

Dear Dr. Halwani,

Thank you for submitting your manuscript to PLOS ONE. After careful consideration, we feel that it has merit but does not fully meet PLOS ONE’s publication criteria as it currently stands. Therefore, we invite you to submit a revised version of the manuscript that addresses the points raised during the review process.

We look forward to receiving your revised manuscript.

Kind regards,

Kota V Ramana, Ph.D.

Academic Editor

PLOS ONE

Journal Requirements:

https://pubmed.ncbi.nlm.nih.gov/40435188/

In your revision ensure you cite all your sources (including your own works), and quote or rephrase any duplicated text outside the methods section. Further consideration is dependent on these concerns being addressed.

Reviewers' comments:

Reviewer's Responses to Questions

**Comments to the Author**

1. Is the manuscript technically sound, and do the data support the conclusions?

Reviewer #1: Partly

Reviewer #2: No

Reviewer #3: Yes

2. Has the statistical analysis been performed appropriately and rigorously?

Reviewer #1: Yes

Reviewer #2: I Don't Know

Reviewer #3: Yes

3. Have the authors made all data underlying the findings in their manuscript fully available?

Reviewer #1: No

Reviewer #2: No

Reviewer #3: Yes

4. Is the manuscript presented in an intelligible fashion and written in standard English?

Reviewer #1: Yes

Reviewer #2: No

Reviewer #3: Yes

Reviewer #1: Manuscript submission:

In the paper entitled “STING Pathway Contributes to Steroid-Hyporesponsive Lung Inflammation in DSS-Induced Mice Model”, the authors study lung inflammation in a commonly utilized mouse model of inflammatory bowel disease (IBD), that is, the DSS colitis model. They perform experiments to test the hypothesis that bacterial translocation and/or DNA activates the STING pathway, which leads to lung inflammation in this model of disease. They first illustrate the pathologic features of their DSS colitis model and confirm evidence of lung inflammation in the model. They then show that bacterial 16s RNA and RNA and protein of several STING pathway components are upregulated in the lungs of the DSS colitis mice. They then examine biomarkers of steroid responsiveness including GR-a, GR-b, and HDAC-2 in lung tissue, finding a signature of reduced steroid responsiveness in the lungs of DSS colitis mice. Finally, they treat mice with either dexamethasone or H151, a STING pathway inhibitor, and remeasure histologic lung inflammation, STING pathway RNA and protein levels, and the steroid responsiveness biomarkers, overall illustrating that treatment with H151 reduces lung inflammation and increases biomarkers of steroid responsiveness as compared to no treatment. They show that dexamethasone also has this impact, although to a lesser extent, although they do not directly compare dexamethasone treated mice to H151-treated mice, or treat mice with dexamethasone and H151 in succession.

Major comments:

- Fig 1 (DSS colitis model characterization) could probably be a supplemental figure, or combined with Fig 2, since the DSS colitis model is so well established in the literature. Fig 1 is not offering anything new in this respect. Moreover, much of the data presented in Figures 1 and 2 are replicated in the authors’ recent manuscript published in PLOS One (“The role of gut leakage and immune cell miss-homing on gut dysbiosis-induced lung inflammation in a DSS mice model”: https://journals.plos.org/plosone/article?id=10.1371/journal.pone.0324230), providing more impetus for moving these figures to supplemental.

- Pertaining to the results presented in Figs 4-5:

o Treatment response should also be assessed by looking at established endpoints, similar measures to Fig 1 and 2 (DAI, H&E, weight, colon length etc).

o Rational for looking at these steroid responsive biomarkers should be presented in section 3.4 – could use a shortened version of lines 410-425 (and remove from Discussion)

- Pertaining to results presented in Fig 5:

o. DSS-dex and DSS-H151 were compared to DSS alone, but DSS-dex should also be compared to DSS-H151.

o The authors claim that STING inhibitor restores steroid responsiveness, so they must show that steroids given with STING inhibition improve histologic and protein endpoints. The STING inhibitor could be given in parallel or sequentially with steroids, but this is a key experiment for the central claim of the manuscript.

o Also, is the GR-a/GR-b ratio also statistically increased in H151 treated mice as compared to DSS and DSS-dex? A plot quantitating the GR-a/GR-b ratio in Figure 5E would help highlight these differences.

- For Supplemental Fig 1A, the authors state that “no noticeable effect” was seen on Sting, Tbk1, Irf3 and IFN-B expression in dexamethasone treated mice; however, all of these had statistically significantly decreased mRNA in dexamethasone treated mice according to p-values presented. The language should be modified, for example “statistically significant but small reduction”.

- Discussion: the authors should not start by stating “This research is the first to demonstrate a direct link between colitis-induced lung inflammation mediated by the STING pathway”. Although the findings they present are suggestive of a connection, they do not directly illustrate that the bacteria translocated from the gut. However, their recent manuscript entitled “The role of gut leakage and immune cell miss-homing on gut dysbiosis-induced lung inflammation in a DSS mice model” (PLOS One) more convincingly illustrates the presence of gut microbes in the lungs of DSS colitis mice. Therefore, the authors should start the discussion by explaining how the findings in their recently published manuscript and the findings in the current manuscript additively support their hypothesis.

- The authors should be careful not to overstate their conclusions. The findings in this manuscript do not prove that the STING pathway was being directly activated by the bacteria. Perhaps using antibiotics or germ-free mice could test this hypothesis, but as the manuscript stands now, a causative link between bacterial translocation and STING pathway activation cannot be established.

Minor comments:

- Line 64: 28-58% seems high. I looked at the cited literature here and this range is indeed based on 3 very small cohort studies (~30 patients per study). This is not in line with clinical experience. I would suggest removing this statistic and in place saying something like “Once considered rare, IBD-associated pulmonary disease is an increasingly reported entity.”

- Lines 140-143: Describe gut and lung inflammation scoring systems (ex, describe the scale 0-4 as shown in Fig 1e and scale 0-3 in 2B) by pathologists in more depth here

- Line 192: Define “DAI” here (reader might have missed it in Methods, should give reminder)

- Lines 253-259: This contains too much conjecture for a Results section. Should be moved to Discussion.

- Line 261: is the word “components” missing after “key STING pathway…”?

- Typo in line 388, should be “increased lung inflammation”

- Line 389: why is gut capitalized?

- Lines 405-408 – should these sentences be combined? The sentence starting with “While the NLRP3…” seems incomplete.

- Discussion should not reference Figures directly (ex. line 427, line 438, line 441).

- Line 434: should be “triggers”

- Line 436: should be “the STING pathway”

Reviewer #2: The study of steroid nonresponse and the gut-lung axis in IBD is of interest, and this manuscript attempts to investigate these phenomena through murine DSS colitis model. The authors focus on pulmonary inflammation and changes in response to dexamethasone and H151, a STING (TMEM173) inhibitor.

However, I have several concerns with regards to the methods and how the text is written/presented:

1) The hypothesis/aim(s) is not clear, especially in the context of the methods and results.

- Last paragraph of introduction discusses the importance of studying STING in IBD-associated lung inflammation, and the potential importance of inhibiting STING to restore steroid responsiveness in IBD. The way this paragraph is written, it is not clear if you are interested in: (A) lung inflammation, or (B) if you are interested in bowel disease, or (C) lung inflammation as an EIM of IBD; all in the context of steroid response. If you are interested in IBD, why not study STING and H151 in the colon of DSS-treated mice? And if you are aiming to study the role of STING in the GLA of IBD, then the introduction needs to be more clear on the background literature. Because my understanding is that if IBD is under control (in remission), then in theory there should be no detrimental GLA/pulmonary inflammation. So the issue is IBD, not pulmonary inflammation.

- My interpretation of the introduction and discussion is that STING inhibition may alleviate steroid hyporesponsiveness/resistance. But you are missing the key experimental group to backup this claim: DSS-treated mice treated with both H151 and dexamethasone. And again, if the background issue (unmet clinical need) is poor response to steroids in IBD patients, then this study needs to be completed in the context of IBD – i.e. the experimental design needs to include gut mucosa.

2) Paragraph structure:

- the first 1-3 sentences for each of the results sections should either be in the introduction or discussion. Results should be limited to results and not discussion around background literature or implication/interpretation of results

- there are critical methods missing from the methods section. E.g. the dosing schedule of H151 is only listed in a figure caption, but not the methods section.

- Page 3, line 83-85 – this paragraph (or its context) is out of place. This discusses steroid nonresponse in IBD, but has nothing to do with the GLA or the study in the paper, which is studying pulmonary inflammation related to IBD and steroid nonresponse in the lungs.

3) STING should be more clearly defined/explained in the introduction… e.g. it is a transmembrane protein, encoded by the STING1 gene (also known as TMEM173).

4) Please provide more details on the 16S rRNA in methods.

- was a DNase step completed?

- what controls were implemented to account for contamination? (especially in context of low biomass tissue, like the lung.)

- in results, 16S is commonly referred in the context of DNA, I assume this is a typo?

5) How do we know that the intranasal administration of dexamethasone or H151 didn’t have their pulmonary effects, not due to inhalation, but rather due to systemic absorption and/or GI tract ingestion (i.e. pulmonary inflammation was reduced because the DSS colitis improved)?

6) Can airway responsiveness (AHR) be affected by systemic or intra-abdominal (colonic) inflammation? Severe weight loss? If so, how were these accounted for in the interpretation of your results?

7) Page 12 – dexamethasone is described as having no noticeable effect, yet the p values are all below 0.05. Please clarify.

Other comments:

1) Several acronyms are either not defined or are defined on their second use. Please ensure that acronyms are defined upon first use (mostly in introduction). e.g. STING, cGAS, CD, UC.

2) Page 3, Line 83: disagree with word choice “mainstay”. We actively try to avoid steroids, and when used, we limit to the shortest possible course and dose. Mainstay treatments are 5-ASA, biologics, and small molecules (+/- thiopurines).

3) Page 3, Line 58: Use of “chronic” implies there is a different disease, “acute” IBD (but there is no such thing). Remove “chronic”.

4) histology scoring was completed by 2 blinded investigators – was there disagreement? How was disagreement resolved?

5) related to comments above, methods state that both lung and colon tissue were used for gene expression, but it looks like only lung tissue results are presented. Is this correct? Either clarify methods or present colon results as well (if relevant).

Reviewer #3: 1.This study innovatively identifies the STING pathway as a core mediator of colitis-induced steroid-hyporesponsive lung inflammation, with a rigorous design, multi-dimensional detection, and confirmation of H151’s superiority over dexamethasone, holding both mechanistic innovation and clinical translational value.

2.The study finds H151 restores the GRα/GRβ ratio and HDAC2 expression but fails to explore how the STING pathway specifically regulates these steroid markers, If possible supplement knockdown/overexpression experiments of key STING molecules and detection of inflammatory pathway activity.

3.The study lacks specific mouse breeding environment parameters, exact weight monitoring frequency, standardized euthanasia procedures, and should supplement sample size calculation basis or increase repeated experiments to enhance result reliability.

4.The study draws conclusions from a mouse model but insufficiently connects to clinical data, requiring supplementary discussion of clinical research progress on the STING pathway in human IBD patients to clarify the link between model conclusions and human diseases.

5.The study meets animal ethics requirements but should supplement mouse distress evaluation criteria in the methods section to improve ethical implementation details.

6.No issues like duplicate publication or conflicts of interest are found, with standardized references and logical data-conclusion consistency, complying with publication ethics.

7.This study has significant scientific value, and addressing the above issues through supplementary data and descriptions will improve its completeness and persuasiveness, making it suitable for publication after revisions.

**Do you want your identity to be public for this peer review?** For information about this choice, including consent withdrawal, please see our For information about this choice, including consent withdrawal, please see our Privacy Policy .

Reviewer #1: No

Reviewer #2: No

Reviewer #3: No

---

## [Author Response · Author response to Decision Letter 1]

12 Jan 2026

All the comments have been addressed in the response to reviewer files and revised manuscript

---

## [Decision Letter · Decision Letter 1]

17 Feb 2026

Dear Dr. Halwani,

Thank you for submitting your manuscript to PLOS ONE. After careful consideration, we feel that it has merit but does not fully meet PLOS ONE’s publication criteria as it currently stands. Therefore, we invite you to submit a revised version of the manuscript that addresses the points raised during the review process.

We look forward to receiving your revised manuscript.

Kind regards,

Kota V Ramana, Ph.D.

Academic Editor

PLOS One

Journal Requirements:

Reviewers' comments:

Reviewer's Responses to Questions

**Comments to the Author**

Reviewer #2: All comments have been addressed

2. Is the manuscript technically sound, and do the data support the conclusions?

Reviewer #2: Partly

3. Has the statistical analysis been performed appropriately and rigorously?

Reviewer #2: Yes

4. Have the authors made all data underlying the findings in their manuscript fully available?

Reviewer #2: Yes

5. Is the manuscript presented in an intelligible fashion and written in standard English?

Reviewer #2: Yes

Reviewer #2: Thank you for addressing previous comments/concerns. The paper and its direction is much clearer now.

I have two remaining questions/concerns:

1) Mice were sacrificed on day 7 or 15, with a total of 25 mice (5 per experimental or control group). There were 5 groups: control, DSS, DSS-Dex, DSS-H151, DSS-Dex-H151. As per figure 1 and 1a, it looks like controls and DSS mice (each N=5) were sacrificed on day 7. But the other groups on day 15? Is this correct? Can you please confirm if you are comparing apples to apples with time courses of treatment and time-lines, clarify as needed.

2) Also, it appears that there are no sham/vehicle controls for the Dex and H151 treatment groups - to negate any effects of the vehicle/conditions of treatment (vs the drug of interest itself). Is there any reason not to include a sham group (both for DSS and for intranasal treatments).

**Do you want your identity to be public for this peer review?** For information about this choice, including consent withdrawal, please see our For information about this choice, including consent withdrawal, please see our Privacy Policy .

Reviewer #2: No

---

## [Author Response · Author response to Decision Letter 2]

18 Feb 2026

1) Mice were sacrificed on day 7 or 15, with a total of 25 mice (5 per experimental or control group). There were 5 groups: control, DSS, DSS-Dex, DSS-H151, DSS-Dex-H151. As per figure 1 and 1a, it looks like controls and DSS mice (each N=5) were sacrificed on day 7. But the other groups on day 15? Is this correct? Can you please confirm if you are comparing apples to apples with time courses of treatment and time-lines, clarify as needed.

Response: We thank the reviewer for raising this important point, the number of mice were not corrected by mistake. Two independent experimental cohorts were used. In cohort 1, control and DSS mice were sacrificed at day 7 to confirm establishment of lung inflammation. In cohort 2, mice were treated from day 1 to 7 and sacrificed at day 15 to evaluate treatment effects. Comparisons were made within the same timepoint only. We have clarified this in the Methods.

2) Also, it appears that there are no sham/vehicle controls for the Dex and H151 treatment groups - to negate any effects of the vehicle/conditions of treatment (vs the drug of interest itself). Is there any reason not to include a sham group (both for DSS and for intranasal treatments).

Response: We apologize for not explicitly clarifying the use of vehicle controls and appreciate the reviewer’s valuable comment.

In the 7-day treatment cohort, control mice received normal drinking water since DSS was given in water. In the 15-day treatment cohort, dexamethasone and H151 were dissolved in PBS and administered intranasally in a total volume of 20 μL per mouse. Control mice in this cohort (15 days) received intranasal PBS (20 μL) following the same schedule as treatment groups. This information has now been clearly detailed in the Methods section.

---

## [Editor Report · Decision Letter 2]

22 Feb 2026

STING Pathway Contributes to Steroid-Hyporesponsive Lung Inflammation in DDS-Induced Colitis Mice Model

PONE-D-25-49308R2

Dear Dr. Halwani,

We’re pleased to inform you that your manuscript has been judged scientifically suitable for publication and will be formally accepted for publication once it meets all outstanding technical requirements.

Kind regards,

Kota V Ramana, Ph.D.

Academic Editor

PLOS One
---

## [Editor Report · Acceptance letter]

PONE-D-25-49308R2

PLOS One

Dear Dr. Halwani,

I'm pleased to inform you that your manuscript has been deemed suitable for publication in PLOS One. Congratulations! Your manuscript is now being handed over to our production team.

Kind regards,

on behalf of

Dr. Kota V Ramana

Academic Editor

PLOS One